# The Antibacterial Effect of Platelets on *Escherichia coli* Strains

**DOI:** 10.3390/biomedicines10071533

**Published:** 2022-06-28

**Authors:** Amina Ezzeroug Ezzraimi, Nadji Hannachi, Antoine Mariotti, Clara Rolland, Anthony Levasseur, Sophie Alexandra Baron, Jean-Marc Rolain, Laurence Camoin-Jau

**Affiliations:** 1Aix Marseille University, IRD, APHM, MEPHI, IHU Méditerranée Infection, 13385 Marseille, France; amina.ezzeroug.ezzraimi@gmail.com (A.E.E.); n_adji07@live.fr (N.H.); antoine-julien.mariotti@ap-hm.fr (A.M.); sophie.baron.2@univ-amu.fr (S.A.B.); jean-marc.rolain@univ-amu.fr (J.-M.R.); 2IHU Méditerranée Infection, Boulevard Jean Moulin, 13385 Marseille, France; rolland.clara@sfr.fr (C.R.); anthony.levasseur@univ-amu.fr (A.L.); 3Département de Pharmacie, Faculté de Médecine, Université Ferhat Abbas Sétif I, Sétif 19000, Algeria; 4Laboratoire d’Hématologie, Hôpital de la Timone, APHM, Boulevard Jean-Moulin, 13385 Marseille, France; 5Aix Marseille University, IRD, SSA, APHM, VITROME, IHU Méditerranée Infection, 13385 Marseille, France

**Keywords:** *Escherichia coli*, platelets, O-antigen, lipopolysaccharide

## Abstract

Platelets play an important role in defense against pathogens; however, the interaction between *Escherichia coli* and platelets has not been well described and detailed. Our goal was to study the interaction between platelets and selected strains of *E. coli* in order to evaluate the antibacterial effect of platelets and to assess bacterial effects on platelet activation. Washed platelets and supernatants of pre-activated platelets were incubated with five clinical colistin-resistant and five laboratory colistin-sensitive strains of *E. coli* in order to study bacterial growth. Platelet activation was measured with flow cytometry by evaluating CD62P expression. To identify the difference in strain behavior toward platelets, a pangenome analysis using Roary and O-antigen serotyping was carried out. Both whole platelets and the supernatant of activated platelets inhibited growth of three laboratory colistin-sensitive strains. In contrast, platelets promoted growth of the other strains. There was a negative correlation between platelet activation and bacterial growth. The Roary results showed no logical clustering to explain the mechanism of platelet resistance. The diversity of the responses might be due to strains of different types of O-antigen. Our results show a bidirectional interaction between platelets and *E. coli* whose expression is dependent on the bacterial strain involved.

## 1. Introduction

Platelets have been widely described as the main actors in hemostasis and thrombosis. More recently, the important role they play in inflammation and defense against pathogens has been highlighted [1,2]. Bacteria interact with platelets through three main mechanisms: (i) by direct binding: bacteria can express surface proteins which allow them to interact directly with surface receptors on platelets and bind to them, as in the case of *Streptococcus sanguinis*, which showed the ability to bind directly to GPIbα [3]; (ii) by binding through plasma proteins: bacterial proteins are capable of binding to fibrinogen and vWF (von Willebrand Factor), serving as a bridge between two cells, as in the example of *Staphylococcus aureus* expressing surface protein A (SpA) that binds to vWf and which in turn binds to platelet GPIbα [4]; (iii) and by binding through the secretion of bacterial products, such as toxins, which interact with platelets and activate them. Among these molecules, the Shiga toxin secreted by *Escherichia coli* induces platelet aggregation by binding to glycosphingolipid receptors on the surface of platelets [5,6,7,8,9,10]. This difference in interaction mechanisms, which is mainly dependent on the bacteria, induces distinct platelet responses. In most cases, this interaction leads to platelet activation, followed by a release of their granular contents composed of, among other things, microbicidal proteins and chemokines that facilitate the destruction of pathogens, signal immune cells and promote inflammation [11,12].

Platelet microbicidal effects have been extensively studied for Gram-positive bacteria. It has been demonstrated that platelets decrease the growth of *Staphylococcus aureus* [13,14,15]. In contrast, this effect has been less studied for Gram-negative bacteria, and such data are scarce. Platelets have been shown to interact with *Escherichia coli* through the platelet Toll-Like Receptor 4 (TLR4) and bacterial lipopolysaccharide (LPS) [5,16,17]. The initiation of the pro-inflammatory signal by LPS depends on the interaction between the TLR4 complex and lipid A, a fragment of LPS [18]. However, other research has shown that platelet activation and aggregation occurs through FcγRIIA without the involvement of TLR4 [19,20]. In contrast, Matus et al. demonstrated that platelet activation is dependent on TLR4 but without FcγRIIA engagement [21]. It is important to note that studies have been carried out to investigate the effects of *E. coli* on the activation and aggregation of platelets, specifically in the case of hemolytic uremic syndrome (HUS), but very few data are available on the effects of platelets on *E. coli*. These studies have tested distinct strains and serotypes and different operating protocols, such as the platelet-bacteria ratio and platelet form [21,22,23].

The aim of this study was to investigate the interaction between platelets and different human *E. coli* strains by initially evaluating the antibacterial effect of platelets, and then evaluating the effect of *E. coli* strains on platelet activation. Moreover, our objective was to compare ten strains of *E coli* having different characteristics, including their colistin sensitivity profile and their pathogenic capability. 

## 2. Materials and Methods

### 2.1. Platelet Preparation

Blood was drawn by venepuncture in sodium citrate from healthy subjects who were not receiving antibiotics, anti-inflammatory, or anti-platelet drugs. Platelet rich plasma (PRP) was prepared according to International Society on Thrombosis and Hemostasis (ISTH) recommendations [24]. A platelet count was performed using a hematology analyzer. PRP was again centrifuged at 1100 g for ten minutes to obtain a platelet pellet that was suspended in phosphate buffered saline (PBS) to obtain a solution of 4 × 10^9^/L. Platelets were then kept at 37 °C in order to prevent activation. The protocol was approved by the ethics committee of the IHU Méditérrannée Infection (Reference 2016–002). All of the subjects gave their written informed consent in accordance with the Declaration of Helsinki.

### 2.2. Bacterial Preparation 

In order to test strains of *E. coli* against platelets and to see if there is a possible cross-resistance between resistance to colistin and resistance to platelet antimicrobial peptides, ten strains of *E. coli* were selected. Five laboratory colistin-sensitive strains were used (ATCC 25922, ATCC 11303, K12, J53 and BL21DE3). Five colistin-resistant human isolates stored at the IHU were also tested (LH1, LH30, Q1065, Q1066 and Q6269) (Table 1). Identification was confirmed using matrix-assisted laser desorption/ionization time-of-flight (MALDI-TOF) mass spectrometry and the Biotyper database (Bruker, Dresden, Germany). Strains were grown at 37 °C in an overnight culture of Columbia agar +5% sheep blood (bioMérieux, Marcy l’Etoile, France). After 18 h of incubation at 37 °C, the colonies were removed and suspended in 0.9% NaCl medium to obtain the required concentrations: 1 × 10^8^ CFU (Colony Forming Unit)/mL for flow cytometry and 3 × 10^8^ CFU/mL for growth test. 

For each strain, the minimum inhibitory concentration (MIC) of colistin (Table 1) was tested by microdilution in accordance with the recommendations of the European Committee on Antimicrobial Susceptibility Testing (EUCAST).

### 2.3. Analysis of Platelet Activation by Flow Cytometry 

The washed platelets were used at a concentration of 2.5 × 10^9^/L, adjusting the concentration with PBS. They were incubated with bacteria in a 1:2.5 bacteria-platelet ratio for one hour at 37 °C and identified by expression of the CD41-FITC antibody (4 μL, IgG, Beckman Coulter, Villepinte, France), as previously described [31]. To determine possible platelet activation following incubation with the bacteria, the expression of the CD62-PC5 antibody (4 μL, IgG, monoclonal, BD sciences, San Jose, CA, USA) on their surface was measured using flow cytometry (Beckman Coulter, FC500, Fullerton, CA, USA). The platelet activator, Thrombin receptor-activating peptide 6 (TRAP) (STAGO®, Asnières, France) (10 μM), and untreated platelets were used as controls. The results were represented by a Mean fluorescence intensity (MFI) percentage of P-Selectin; the expression was calculated using the following equation: (MFI of platelets incubated with bacteria) × 100/(MFI of platelets alone).

### 2.4. Platelet Supernatant Effect on Bacterial Growth 

To obtain the supernatant from the activated platelets, the activated platelets were incubated with the J53 strain for one hour at 37 °C. The mixture underwent three successive centrifugation rounds (1300× *g*, 5000× *g* and 5000× *g*; ten minutes each) and the supernatant was recovered and filtered (0.22 μm), then incubated with bacteria for four hours at 37 °C. Mixtures were serially diluted as described above, then spread on agar and the colonies were counted the following day. 

### 2.5. Pangenome Analysis

The genomes of the seven strains of *Escherichia coli* (ATCC25922, ATCC11303, K12, J53, BL21 DE3, LH1 and LH30) were retrieved from the National Centre for Biotechnology Information (NCBI) database. For the three other strains, Q1065 (unpublished data), Q1066 (unpublished data) and Q6269 (JAIBLN000000000), the genomes were obtained from the IHU sequencing platform by Illumina MiSeq according to a paired-end strategy. 

The first step in performing the pangenome was to predict the ORFs (Open Reading Frame) for each strain with PROKKA software using the default parameters [32]. Then, Roary software was used to build the pangenome with the core genome alignment default parameters [33]. A graphic representation of the pangenome results was prepared using the roary_plots.py script provided on the Roary website.

### 2.6. O-antigen Strain Serotyping

The serotyping of the *E. coli* strains (Table 1) used in this study was performed in silico from the genomes obtained by high-throughput sequencing. The FASTA sequences of these genomes were analyzed using ECTyper software [34], which makes it possible to serotype the O and H antigens of *E. coli* and *Shigella* spp. The bioinformatic predictions made with this software were then compared with those obtained by the Serotype Finder 2.0 prediction module of the Centre for Genomic Epidemiology developed by the Technical University of Denmark [35]. The results of all of the strains were concordant after analysis by both databases. 

### 2.7. Statistical Analysis

Statistical analyses were performed using GraphPad Prism 9 for Windows (GraphPad Software, La Jolla, CA, USA). Significant differences (for bacterial growth and flow cytometry) between the two groups were determined using the two-tailed, paired Student’s *t*-test. The statistical significance was set at *p* < 0.05. A test of normality was applied on the effect of platelets on bacterial growth data, and it turned out that samples (negative control and platelets + bacteria for each strain) show a Gaussian distribution following a verification by the Shapiro-Wilk test (*p* > 0.05).

While for supernatants effect, they were determined using the Bonferroni’s multiple comparisons test precede by Two-way ANOVA test, considering *p* < 0.05 as statistically significant. The correlation between bacterial growth and MFI percentage values was determined using the Pearson test.

## 3. Results

### 3.1. The Effect of Platelets on the Growth of E. coli Strains

After four hours of incubation, the platelets significantly decreased bacterial growth of three *E. coli* laboratory strains, as compared to the controls (*n* = 5 for each strain: *p* = 0.0036, <0.0001 and 0.0017 for ATCC11303, BL21DE3 and J53, respectively, paired Student’s *t*-test) (Figure 1). These strains were referred to as “platelet sensitive”. 

In contrast, the platelets significantly promoted the growth of *E. coli* K12 and *E. coli* ATCC25922 (*n* = 5 for each strain: *p* = 0.0013 and 0.0065, respectively). Likewise, the growth of all the clinical strains was enhanced with platelets (*n* = 5 for each strain: *p* = 0.0001, 0.0005, 0.0105, 0.0009 and 0.0079, for LH1, LH30, Q1065, Q1066, and Q6269, respectively, paired Student’s *t*-test) (Figure 1). These seven strains were referred to as “platelet resistant”. 

### 3.2. The Effect of Platelet Supernatant on the Growth of E. coli Strains 

Mixes of the supernatants of platelets previously stimulated by the *E. coli* J53 strain were prepared, then re-incubated again with bacteria, as described in the Section 2. The supernatant of the platelets treated by TRAP was used as a positive control.

The supernatants showed effects similar to whole platelets on the growth of tested strains. Indeed, supernatants of platelets stimulated with *E. coli* J53 and TRAP significantly inhibited the growth of *E. coli* BL21DE3 and *E. coli* ATCC 11303 strains ((*n* = 5: *p*= 0.0223, 0.0293 and *p* = 0.0381, 0.0048 for BL21 and J53, respectively, Figure 2A). In contrast, the supernatants of platelets stimulated with *E. coli* J53 and TRAP continued to promote the growth of K12 (*n* = 5: *p* = 0.0203 and 0.0332, Figure 2B). Q1065 growth was also enhanced by supernatants of platelets stimulated with *E. coli* J53 (*n* = 5: *p* = 0.0233). On the other hand, platelet supernatants had no significant effect on ATCC 25922 growth. 

### 3.3. The Effect of E. coli Strains on Platelet Activation

In order to assess whether *E. coli* strains induce platelet activation, P-selectin was measured using flow cytometry. After one hour of co-incubation, five strains (ATCC11303, J53, BL21DE3, K12 and ATCC25922) significantly increased P-selectin compared to the controls (platelets alone). All of the other strains tested did not increase the expression of P-selectin. Significant differences between platelets alone and platelets treated by TRAP as well as between platelets alone and platelets infected with bacteria were determined using the two-tailed, paired Student’s *t*-test. Statistical significance was set at *p* < 0.05 (Table 2). 

The Pearson test was applied to test the correlation between colony count values and the MFI percentage of the different strains in co-incubation with platelets. A negative significant correlation was obtained (Pearson r = −0.6795, *p*-value = 0.0307).

## 4. Pangenome Analysis

In order to better understand the difference in platelet activation between the *E. coli* strains as previously shown, a pangenome analysis was performed (Figure 3). This study was designed to identify the difference between platelet-activating and non-platelet-activating strains. We were particularly interested in genes present in only one or the other groups of bacteria. In the non-platelet-activating group that consisted of *E. coli* strains LH1, LH30, Q1065, Q1066 and Q6269, no genes shared by all strains were found. In contrast, in the platelet-activating group, six common genes were identified. Of them, two were annotated as elongation factor Tu (elongation factor Tu 1 and 2), two others were annotated as transposase (IS3 family transposase IS911, IS4 family transposase IS4), another was annotated as Outer membrane porin protein OmpD, and the last was annotated as a lactose phosphotransferase system repressor. None of these functions showed a direct link to platelet activation. Furthermore, the analysis of the classification of strains following the pangenome study did not indicate a clustering of bacteria according to platelet activation. Indeed, the clustering put *E. coli* strains BL21 and ATCC11303 in a first group and *E. coli* strains J53, K12 and ATCC25922 in the second (Figure 3). 

Thus, the bioinformatic analyses did not identify a gene or a cluster of genes as being at the origin of the difference in platelet activation. 

## 5. O-antigen Strain Serotyping

We investigated the difference in behavior between two genetically related strains, K12 and its mutant strain J53. Using the Genome Mapper of the EcoCyc database (ecocyc.org) and searching for genes coding for proteins involved in the biosynthesis of the O antigen, and thus in the formation of LPS, we found that the *wbbL* gene had an IS5 insertion in its sequence, making it non-functional, and thus able to modify the structure of LPS (Figure 4).

## 6. Discussion

In this study, we evaluated the consequences of interactions between platelets and ten strains of *Escherichia coli*. Our results showed that platelets had an antibacterial activity on three laboratory strains among the ten tested. Genomic comparison of two strains with different susceptibility profiles revealed a difference in the genomic cluster coding for the O-antigen.

To our knowledge, few studies have evaluated the bactericidal effect of platelets against *Escherichia coli* strains [19,36]. Moreover, the originality of our study resides in the fact that we tested a large panel of strains, including both laboratory and clinical strains, which are further distinguished by their colistin-resistance profile.

The bactericidal effect of platelets has been previously tested in two studies, where two of the five colistin-sensitive laboratory strains selected in our study were used. Our results confirmed that *E. coli* ATCC1130 growth inhibition was inhibited by platelets, as previously demonstrated by Tohidnezhad et al. [37]. Moreover, as described by Cieślik-Bielecka et al., we confirmed that the growth of the *E. coli* ATCC25922 strain was not inhibited by platelets [38]. The concordance of these results validated the choice of our experimental model. 

Among the ten strains tested, only three were sensitive to platelet bactericidal activity. The interaction of platelets with these strains induced platelet activation responsible for a secretion process of platelet granule content, as evidenced by the increased expression of P-selectin. The bactericidal activity observed is probably due to the action of microbicidal platelet peptides released, since the same effect is observed with the supernatant of activated platelets. We have already described this mechanism for *Staphylococcus aureus* [13]. These three strains of *E. coli*, which were sensitive to platelets, are all laboratory strains that do not express resistance mutations and are not responsible for human infectious pathologies. Interestingly, the two other *E. coli* laboratory strains tested (K12 and ATCC 25922) also induced platelet activation. Regarding this last point, our results are in line with those of Fejes et al., who also demonstrated that the K12 strain induces an increase in P-selectin and CD63 [22]. However, they are insensitive to platelet bactericidal activity. This lack of effect could be the consequence of weak platelet activation induced by these strains, as we have demonstrated a negative correlation between the inhibitory effect of platelets and the activating effect of *E. coli* strains, which means the more the strains increase platelet granule release, the less bacterial growth decreases in the presence of platelets. However, regarding the *E. coli* K12 strain, Palankar et al. found that the bactericidal effect was only obtained with the LPS mutant *E. coli* strains, but not on the wild strain (K12) [36].

Previous studies have suggested that the difference between the profiles of the *E. coli* strains regarding platelets can be explained by the existence of two types of LPS and their interactions with immune cells, which may be the same mechanism for platelets [22]. Indeed, “rough” LPS could activate a wider range of cells with greater efficiency compared to the “smooth” form [39]. Macrophages have been shown to be able to respond to “rough” LPS and lipid A, but not to “smooth” LPS. Furthermore, the “smooth” form requires CD14 to activate immune cells [40]. 

When looking at the overall effect of each strain on the different parameters studied (platelet activation and inhibitory effect), different platelet interaction profiles can be determined. It could be hypothesized that these profiles are dependent on the structure of the LPS O-antigen of each *E. coli* strain. Indeed, it has long been shown that platelets expose TLR-4 on their surface, which is involved in the recognition of LPS [41]. It could be hypothesized that structural abnormalities of LPS might induce an alteration of the phenomenon of recognition of bacterial structural patterns through this TLR-4. 

Resistance to colistin can also be implicated in generating a difference in responses to platelets, since LPS more precisely lipid A, represents the target of colistin, which is also the principal element that interacts with the platelet receptor TLR4. A possible modification of LPS can cause a defect in the interaction with platelets, which can lead to platelet non-activation and resistance to platelet peptides, especially cationic ones which share several characteristics with colistin, namely their polarity and their modes of action [42,43]. Moreover, the 5 colistin resistant strains could not activate the platelets and they are all resistant to the platelets, which can constitute a problem during an *E. coli* infection which should not be neglected.

Conclusions could not be drawn from the pangenome results as to the gene(s) responsible for platelet activation or strain sensitivity toward platelets. We therefore turned to the prediction of the O-antigen type and were mainly interested in the comparison between K12 and its mutant J53, which have distinct profiles based on bacterial growth results. The *E. coli* K12 strain is known to lack O-antigen, secondary to the presence of mutations, including an IS5 insertion (Figure 3) in the gene cluster involved in O-antigen biosynthesis, as well as core LPS [44]. From this reference K-12 strain, a mutant was developed (K-12 W3110) by transposing the *rfb* gene cluster from the WG1strain. This K-12 W3110 strain was shown to express an O16-type O-antigen [45]. The J53 strain, which is derived from K12, has a deletion of IS5 in this gene cluster, which may indicate that this J53 strain has a functional O-antigen [46]. Our data, as shown above, demonstrate that the K12 and J53 strains seem to have an opposite profile in terms of platelet bactericidal effect. This could be explained by the changes in the structure of the LPS, which is support by data from the genetic database. This hypothesis, that platelet activation is dependent upon the O-antigen carried by bacteria, can also be supported by the fact that TLR-4-dependent signaling pathways leading to platelet activation and aggregation have been shown to exist [47].

In summary, based on data from the literature and our results, we hypothesize that the platelet activation and the antibacterial effect against *E.coli* originate from the same mechanism of action, potentially the bacterial LPS, and that a variability or a structural modification of the LPS, leads to both a defect in platelet activation and resistance to platelet peptides [48]. We did not confirm the hypothesis that this difference in behavior could be related to colistin resistance, because 2 colistin sensitive strains have the same profile as colistin resistant strains but we are convinced that this notion of cross-resistance should be more emphasized.

We believe that these new observations are worth sharing. However, we are aware that further studies, involving proteomic and genomic analyses, are needed to better explain the molecular basis of the differential behavior of *E. coli* strains towards platelets. 

## 7. Conclusions

In conclusion, our work evaluated the bactericidal effect of platelets on ten *E. coli* strains with different characteristics (clinical strains/laboratory strains, colistin resistance profile). On one hand, we have demonstrated a correlation between platelet activation induced by *E coli* and bactericidal activity. On the other hand, our preliminary data, obtained by studying the structure of the O antigen of two laboratory strains, suggest that modification of the O antigen would be responsible for this sensitivity to the bactericidal activity of platelets.

Since researchers have described the interaction between platelets and *E. coli* as a complex interaction, and the fact that all of the clinical strains tested in our study showed resistance to platelet peptides and that their bacterial growth is increased in the presence of platelets, it is important to further investigate the mechanisms of this interaction. Further work should be conducted by testing other clinical strains and by targeting other phenomena such as platelet aggregation to fully understand and identify all the factors involved in this interaction. This will be necessary in order to establish a clinical model of sepsis and HUS for subsequent optimal use of existing drugs and possible development of new drugs.

## Figures and Tables

**Figure 1 biomedicines-10-01533-f001:**
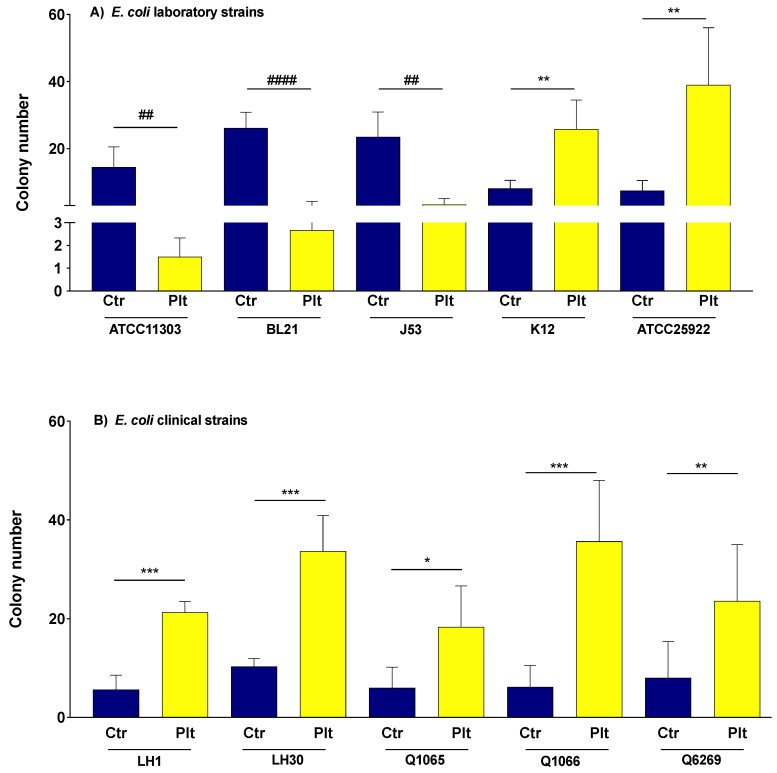
The effect of platelets on the growth of *E. coli* strains. Bacteria (20 μL, 3 × 10^8^ CFU) were added to platelets (180 μL, 4 × 10^9^/L) and incubated together at 37 °C for four hours while being rotated. Bars represent Mean with SD. Ctr = control: bacteria alone (blue column). Plt = bacteria-platelet mixture 1:10 ratio (yellow column). *: significant increase; #: significant decrease. Significant differences between the two groups were determined using the two-tailed, paired Student’s *t*-test. *: *p* < 0.05, ** and ##: *p* < 0.01, ***: *p* < 0.001, ####: *p* < 0.0001.

**Figure 2 biomedicines-10-01533-f002:**
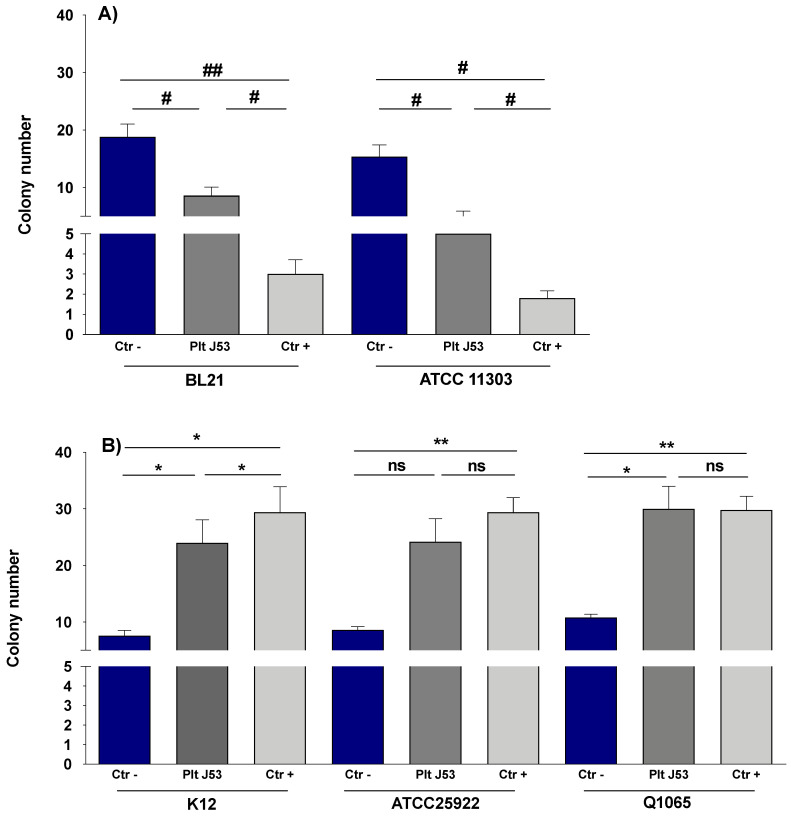
Effect of platelet supernatant on the growth of *E. coli* strains. Bacteria (20 μL, 3 × 10^8^ CFU) were added to filtered supernatant (180 μL) and incubated together at 37 °C for four hours while being rotated. (**A**) Supernatant effect of *E. coli* platelet-sensitive strains BL21DE3 and ATCC11303 (*n* = 5). (**B**) Supernatant effect of *E. coli* platelet-resistant strains K12, ATCC 25922 and Q1065 (*n* = 5). Ctr−: bacteria alone (blue column); Plt J53: bacteria incubated with supernatant of platelets stimulated by J53 (dark grey column); Ctr+: bacteria incubated with supernatant of platelets treated by TRAP (light grey column). Bars represent Mean with SD. *: significant increase; #: significant decrease, ns: non-significant. Significant differences between the two groups were determined by Bonferroni test preceded by two-way ANOVA. * and #: *p* < 0.05, ** and ##: *p* < 0.01, ns: *p* > 0.05.

**Figure 3 biomedicines-10-01533-f003:**
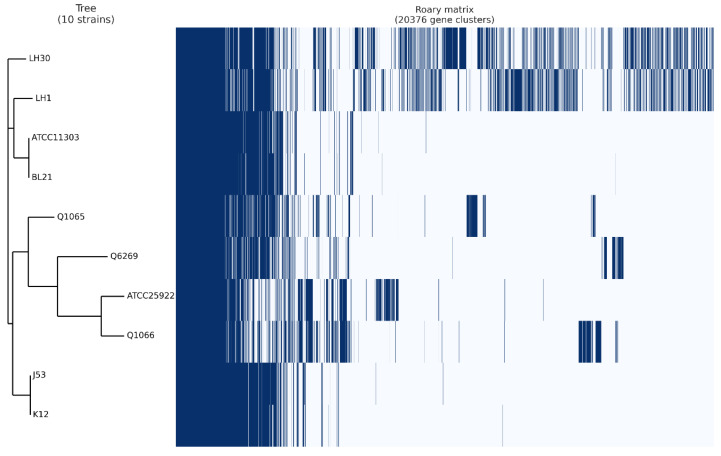
Visualization of pangenome analysis by Roary software of 10 *E. coli* strains. Pangenome analysis of the ten *E. coli* strains using Roary software. Whole genomes of the strains were clustered according to the presence/absence of core genes. Blue: presence of gene, white: absence of gene.

**Figure 4 biomedicines-10-01533-f004:**
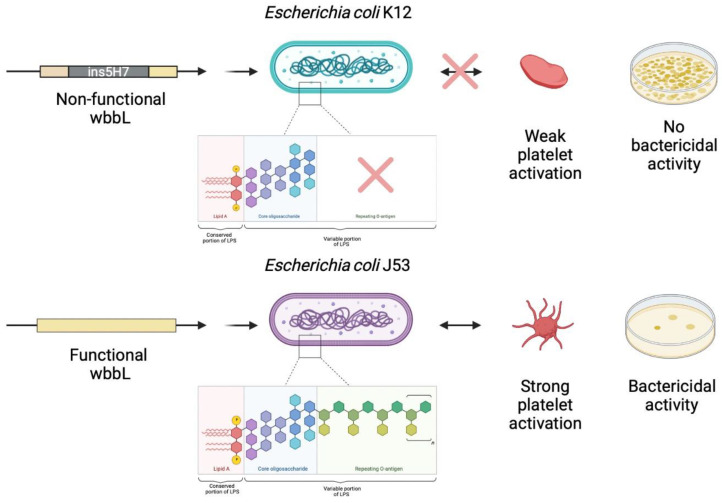
Comparison between the O-antigen biosynthesis cluster of K12 and J53. Comparison between the O-antigen biosynthesis cluster of K12 and its mutant J53. The *E. coli* K12 strain lacks O-antigen, secondary to the presence of mutations, including an IS5 insertion in the gene cluster involved in O-antigen biosynthesis (created with BioRender.com, accessed on 27 April 2022).

**Table 1 biomedicines-10-01533-t001:** Origins and characteristics of *E. coli* strains used in this study.

*Escherichia coli* Strain	Origin	O-Antigen Type	Colistin Resistance Mechanism	MIC	References
IHU clinical isolates					
LH 1	Human	O174	*mcr-1* gene	7.8 mg/L	[25]
LH 30	Human	O8	*mcr-1* gene	3.9 mg/L	[25]
Q1066	Human	O25	Unknow mechanism	7.8 mg/L	Unpublished
Q1065	Human (Pharyngeal swab)	O9	Unknow mechanism	3.9 mg/L	Unpublished
Q6269	Human (urine)	O175	Unknow mechanism	3.9 mg/L	Unpublished
Laboratory strains					
ATCC 25922	Reference strain	O6	-	0.97 mg/L	[26]
ATCC 11303	Reference strain	O7	-	0.48 mg/L	[27]
K12	Human	-	-	1.95 mg/L	[28]
J53	Laboratory mutant of K12	O16	-	0.97 mg/L	[29]
BL 21 DE3	Laboratory mutant of K12	O7	-	0.97 mg/L	[30]

**Table 2 biomedicines-10-01533-t002:** Mean Fluorescence Intensity (MFI) percentage of P-selectin expression of platelets infected with *E. coli* strains.

Mean ± SD of MFI % of Platelets Stimulated with *E. coli* Strains and TRAP			
*E. coli* Strains	Mean SD of Plt-*E. coli*	*p*-Value Plt-*E. coli* Compared to Plt	*p*-Value Summary of Plt-TRAP Compared to Plt
ATCC11303	124.9 ± 14.3	0.007	**
J53	190.7 ± 40.5	0.017	**
BL21DE3	134.5 ± 22.5	0.026	**
K12	106.3 ± 4	0.024	**
ATCC25922	109.1 ± 5.4	0.019	**
LH1	104.2 ± 17.9	-	**
LH30	96.5 ± 7.4	-	**
Q1065	99.4 ± 9.4	-	*
Q1066	102.3 ± 7.8	-	*
Q6269	100.4 ± 9	-	*

Plt: Platelets alone (100%). Plt-*E. coli*: platelets stimulated with *E. coli* strains. Plt-TRAP: platelets treated by TRAP. Percentage of P-Selectin expression in treated and stimulated platelets was calculated by the following equation: (MFI of platelets infected with bacteria) × 100/(MFI of platelets alone).*: *p* < 0.05, **: *p* < 0.01.

## Data Availability

The data that support the findings of this study are available from the corresponding author upon reasonable request.

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
