# Peer review of "The Antibacterial Effect of Platelets on Escherichia coli Strains"

_biomedicines, 2022, doi:10.3390/biomedicines10071533_

Round 1
Reviewer 1 Report
The authors investigate the two-way interaction between platelets
and different human E. coli strains, evaluating the antibacterial effect of platelets, and then evaluating the effect of strains on platelet activation.
It is an interesting idea and I congratulate the authors for it.
In the Introduction section a more detailed description of study motivation and benefits should be described.
I recommend Figures 1 and 2 to be in colour not in black and white
Why are Tables and Figures inserted at the end of the manuscript and not in the manuscript as MDPI requires ?
Figures and Tables should be inserted in the text where are mentioned
There is a Discussion section but not a Conclussion. I recommend a Conclussion section in which the authors can discuss perspectives
Check abbreviation, define all of them
There are quite a few grammar and syntax errors in the text. Please correct it carefully
Author Response
Prof. Laurence Camoin-Jau
Corresponding author
Service d’Hématologie Biologique
Assistance Publique Hôpitaux de Marseille
Centre Hospitalier Universitaire Timone
164 Rue Saint Pierre
13005 Marseille
France
Email: Laurence.camoin@ap-hm.fr
Marseille, 24 May 2022
Mr Conley ChenAssistant EditorMDPI Biomedicines Editorial Office St. Alban-Anlage 66, 4052 Basel, Switzerland
Dear Mr Chen,
We thank you for the critical reading of our work entitled "The antibacterial effect of platelets on Escherichia coli strains” (ID: biomedicines-1726005).
We have made corrections according to the referees' comments. We have responded to all the reviewers' requests.
We kindly ask you to read the corrected version. We sincerely hope that we have answered all questions. We are sending you version with the corrections made in red.
Sincerely yours,
Prof. L Camoin-Jau
Corresponding author
Reviewer 1
Comments and Suggestions for Authors
The authors investigate the two-way interaction between platelets and different human E. coli strains, evaluating the antibacterial effect of platelets, and then evaluating the effect of strains on platelet activation.
It is an interesting idea and I congratulate the authors for it.
Dear reviewer,
We thank you for this favourable and encouraging comment.
1 : In the Introduction section a more detailed description of study motivation and benefits should be described.
Dear reviewer,
We note your remark. A more detailed description of study motivation was described: line 53-56 page 2 ; Check new version.
2/ I recommend Figures 1 and 2 to be in colour not in black and white
As requested, Figures 1 and 2 are in colour. Please check the new version.
3/ Why are Tables and Figures inserted at the end of the manuscript and not in the manuscript as MDPI requires ? Figures and Tables should be inserted in the text where are mentioned
As requested, figures and tables are inserted in the text.
4/ There is a Discussion section but not a Conclusion. I recommend a Conclusion section in which the authors can discuss perspectives
A conclusion was added: page 16 line 324-338.
5/ Check abbreviation, define all of them
All abbreviations were checked. Please check the new version.
6/ There are quite a few grammar and syntax errors in the text. Please correct it carefully
Grammar and syntax errors were corrected

Reviewer 2 Report
The aim of the study by Ezzraimi et. al. was to evaluate the bidirectional interaction between platelets and E. coli strains. It is well known that, although blood platelets are not considered as part of the classical immune system, they have many features indicating their important role in the anti--infective defense. Therefore, presented study seems to be interesting. However, the authors pointed that relatively little is known about platelet interaction with Gram-negative bacteria and that they...."are the first researchers to evaluate the consequences of interactions between platelets and ten strains of Escherichia coli" (L 185-186). I personally disagree with this statement. The authors should particularly emphasize the new approach to the problem and underline the obtained original information that goes beyond the data available for a long time. By assessing the value of the obtained data in each new publication, we expect to present more universal knowledge than is available so far and to explain / justify the more general purpose of the research undertaken.
The authors should consider the following specific comments.
Abstract
- L 15-16 - ...this phrase should be rewrited: "Our goal was to study the two-way interaction between platelets isolated from blood samples of healthy donors and selected strains of Escherichia coli, in order to"......
- L 16-19 ..."Washed platelets were incubated with E. coli strains to study the inhibitory effect of platelets on bacteria. CD62P surface exposure was evaluated by flow cytometry to assess the strain effect on platelet activation. The effect of supernatants from pre-activated platelets on bacterial growth was also assessed". Please connect these underlined information.
- L 21-22 - "Within 10 strains tested, the platelets were able to inhibit the growth of only three laboratory strains, which belong to the group of laboratory strains":.. This sentence should be corrected.
-L 24-26 - This conclusion is not very original - it is rather obvious that 10 randomly selected strains from the group of more than 180 E. coli strains differing in the types of O-antigen, they behave differently in the adopted experimental setup.
Introduction Generally speaking, the Introduction content should be rethought and reorganized. It is currently chaotic. It presents and justifies the purpose of research in very general terms. There is no so-called "fluidity" of the text. Also missing is the so-called significant keynote of the study.
- L 32, 36 - references cited should be combined in the line 36.
- L 32-37 - "Indeed, bacteria interact with platelets through three main mechanisms, by direct binding via receptor complementarity, by binding through plasma proteins serving as a bridge between the two cells, and by binding through the secretion of bacterial products, such as toxins, which interact with platelets. This difference in interaction mechanisms, which is mainly dependent on the bacteria, induces distinct platelet responses"..... This phrase should be clarified. This is too big mental shortcuts that simplify the context.
- L 42-44 - These statements are half-truths. Studies on the interaction of platelets and, for example, S. aureus, date far earlier than the cited work, and studies of the interaction of Gram-positive and Gram-negative bacteria with platelets have been carried out for many years.
Material and methods section. The description of the materials and the methodology used is generally correct.
However, no rationale for selecting these particular 10 E. coli strains is given here (L 68-70) or elsewhere in the manuscript.
- L 63 - ..... 4.108 e/mL (?)
- L 74 - in ..... 0.9 ‰ NaCl medium (?)
- L 75 - ... 1 or 3 x 108 CFU/ mL (?)
- L 80 - .....of 2.5 x 109/L (/mL or /L) (?)
Results section. The data presented in Table 2 and Figures 1 and 2 document the results of a basic experiment on two selected parameters for the interactions between platelets and bacteria.
I have no objections to their presentation. They just reflect what has been assessed.
Discussion section. This part contains many repeats of the description of results. However, in many places the interpretation of the observations made is missing or are too vague or over interpreted. It should be rewrite.
I appreciate the use by the authors of modern molecular methods for antigen typing of the studied E. coli strains. It is now known that there are currently ∼186 different E. coli O-groups and 53 H-types. There are also many pathogenic groups of E. coli that cause disease in humans and animals, including diarrheagenic E. coli and the extra-intestinal pathogenic E. coli that cause illness outside of the GI-tract.
In light of it, I do not see in MS a "key" for the selection of these 10 E. coli strains to explain / establish the original pattern of platelet / bacteria interaction. It would be worthwhile to plan more targeted studies in the context of E. coli infections in which, in the course of pathogenesis, bacteria enter into direct (cell / cell) or indirect (LPS and other bacterial products contact with platelets (sepsis, uro-sepsis, other multi-organ infections). For the time being, at least it would be worthwhile to conduct the discussion like this.
- L 190-198 - the authors themselves admit that the data received did not provide information other than those previously published by others.
- L 188-189 - ..."We hypothesise that the lack of an antibacterial effect of platelets on the other 188 strains tested is due to a modification of the O-antigen". It can be suggested that the Authors should delimit their reflections only to data obtained. In this case only two strains were analyzed in details: K12 and J53 (L-178-179).
They conclude that:..."Taken together, the bactericidal effect of the platelets is therefore dependent on the strain of E. coli used"... ( L-206-207). Assumptions that go far beyond experimental evidence are unfounded.
- L 211-214 - ..."Although we do not have enough data, it could be hypothesised that platelets may represent an unavoidable barrier during E. coli bacteraemia and that the damage that occurs requires further re sistance to this barrier":.. It is not clear the author's hypothetical statement. Such requires a more careful analysis. Since there is no hard experimental evidence for this, such loose speculation is inadvisable.
Author Response
Prof. Laurence Camoin-Jau
Corresponding author
Service d’Hématologie Biologique
Assistance Publique Hôpitaux de Marseille
Centre Hospitalier Universitaire Timone
164 Rue Saint Pierre
13005 Marseille
France
Email: Laurence.camoin@ap-hm.fr
Marseille, 24 May 2022
Mr Conley ChenAssistant EditorMDPI Biomedicines Editorial Office St. Alban-Anlage 66, 4052 Basel, Switzerland
Dear Mr Chen,
We thank you for the critical reading of our work entitled "The antibacterial effect of platelets on Escherichia coli strains” (ID: biomedicines-1726005).
We have made corrections according to the referees' comments. We have responded to all the reviewers' requests.
We kindly ask you to read the corrected version. We sincerely hope that we have answered all questions. We are sending you 2 a version with the corrections made in red.
Sincerely yours,
Prof. L Camoin-Jau
Corresponding author
Reviewer 2:
The aim of the study by Ezzraimi et. al. was to evaluate the bidirectional interaction between platelets and E. coli strains. It is well known that, although blood platelets are not considered as part of the classical immune system, they have many features indicating their important role in the anti--infective defense. Therefore, presented study seems to be interesting.
However, the authors pointed that relatively little is known about platelet interaction with Gram-negative bacteria and that they...."are the first researchers to evaluate the consequences of interactions between platelets and ten strains of Escherichia coli" (L 185-186). I personally disagree with this statement.
Dear reviewer,
Thank you for your comment. We agree that a few studies have already been published on the interactions between E coli and platelets. However, our study is different because we studied 10 strains with different profiles, including their resistance to colistin. We also compared clinical and laboratory strains. To our knowledge, no study has been conducted on such a large panel of strains. At your request, we have deleted this sentence. Please check the new version.
The authors should particularly emphasize the new approach to the problem and underline the obtained original information that goes beyond the data available for a long time.
By assessing the value of the obtained data in each new publication, we expect to present more universal knowledge than is available so far and to explain / justify the more general purpose of the research undertaken.
Dear reviewer,
Thank you for your comment. As requested, we rewrote the discussion to highlight the strengths of our work
The authors should consider the following specific comments.
Abstract
- L 15-16 - ...this phrase should be rewrited: "Our goal was to study the two-way interaction between platelets isolated from blood samples of healthy donors and selected strains of Escherichia coli, in order to"......
- L 16-19 ..."Washed platelets were incubated with E. coli strains to study the inhibitory effect of platelets on bacteria. CD62P surface exposure was evaluated by flow cytometry to assess the strain effect on platelet activation. The effect of supernatants from pre-activated platelets on bacterial growth was also assessed". Please connect these underlined information.
- L 21-22 - "Within 10 strains tested, the platelets were able to inhibit the growth of only three laboratory strains, which belong to the group of laboratory strains":.. This sentence should be corrected.
-L 24-26 - This conclusion is not very original - it is rather obvious that 10 randomly selected strains from the group of more than 180 E. coli strains differing in the types of O-antigen, they behave differently in the adopted experimental setup.
Dear reviewer,
We note all your remarks. We rewrote the abstract. Please check the new version.
Introduction
Generally speaking, the Introduction content should be rethought and reorganized. It is currently chaotic. It presents and justifies the purpose of research in very general terms. There is no so-called "fluidity" of the text. Also missing is the so-called significant keynote of the study.
- L 32, 36- references cited should be combined in the line 36.
Corrections were made
- L 32-37- "Indeed, bacteria interact with platelets through three main mechanisms, by direct binding via receptor complementarity, by binding through plasma proteins serving as a bridge between the two cells, and by binding through the secretion of bacterial products, such as toxins, which interact with platelets. This difference in interaction mechanisms, which is mainly dependent on the bacteria, induces distinct platelet responses"..... This phrase should be clarified. This is too big mental shortcuts that simplify the context.
As requested, this paragraph was clarified. Please check line 23-32 Page 3.
- L 42-44- These statements are half-truths. Studies on the interaction of platelets and, for example, aureus, date far earlier than the cited work, and studies of the interaction of Gram-positive and Gram-negative bacteria with platelets have been carried out for many years.
We agree. We have added new references.
Material and methods section. The description of the materials and the methodology used is generally correct.
However, no rationale for selecting these particular 10 E. coli strains is given here (L 68-70) or elsewhere in the manuscript.
As requested, we have specified the choice of the ten strains.
- L 63 - ..... 4.108 e/mL (?)
- L 74 - in ..... 0.9 ‰ NaCl medium (?)
- L 75 - ... 1 or 3 x 108 CFU/ mL (?)
- L 80 - .....of 2.5 x 109/L (/mL or /L) (?)
All these points have been corrected . Please check page 4-5.
Results section. The data presented in Table 2 and Figures 1 and 2 document the results of a basic experiment on two selected parameters for the interactions between platelets and bacteria.
I have no objections to their presentation. They just reflect what has been assessed.
Dear reviewer,
We thank you for this favourable comment
Discussion section.
This part contains many repeats of the description of results. However, in many places the interpretation of the observations made is missing or are too vague or over interpreted. It should be rewrite.
Dear reviewer,
As you requested, we are rewriting the discussion, taking into account your remarks; we thank you for your favourable comment on our scientific approach
I appreciate the use by the authors of modern molecular methods for antigen typing of the studied E. coli strains. It is now known that there are currently ∼186 different E. coli O-groups and 53 H-types. There are also many pathogenic groups of E. coli that cause disease in humans and animals, including diarrheagenic E. coli and the extra-intestinal pathogenic E. coli that cause illness outside of the GI-tract.
In light of it, I do not see in MS a "key" for the selection of these 10 E. coli strains to explain / establish the original pattern of platelet / bacteria interaction. It would be worthwhile to plan more targeted studies in the context of E. coli infections in which, in the course of pathogenesis, bacteria enter into direct (cell / cell) or indirect (LPS and other bacterial products contact with platelets (sepsis, uro-sepsis, other multi-organ infections). For the time being, at least it would be worthwhile to conduct the discussion like this.
- L 190-198- the authors themselves admit that the data received did not provide information other than those previously published by others.
Response:
- Dear reviewer, we agree that our results are in accord with previous studies. However, the originality of our study lies in the fact that we tested a number of strains at the same time (10). Thus, we have highlighted the strain-dependent character of both bacterial sensitivity to platelets and the activating power of bacteria on platelets. We have shown a correlation between these two effects via the Pearson test. We have compared our results on the anti-bacterial effect of platelets with the sensitivity of bacteria to colistin, involving LPS in its mechanism of action. Furthermore, the use of the supernatant of activated platelets shows a secretory phenomenon of the antibacterial action of platelets. Finally, we added a bioinformatics analysis based on genomic comparison to explain this difference.
- L 188-189- ..."We hypothesise that the lack of an antibacterial effect of platelets on the other 188 strains tested is due to a modification of the O-antigen". It can be suggested that the Authors should delimit their reflections only to data obtained. In this case only two strains were analyzed in details: K12 and J53 (L-178-179).
- We agree that we cannot generalise based on the comparison of genomic profiles of two strains. As requested, we have corrected this sentence. Please check the new submitted version.
They conclude that:..."Taken together, the bactericidal effect of the platelets is therefore dependent on the strain of E. coli used"... ( L-206-207). Assumptions that go far beyond experimental evidence are unfounded.
We assessed the antibacterial effect of platelets on 10 strains. The latter showed different sensitivity. Also, some previous studies have reported this observation separately. Thus, we can conclude that this effect depends on the E. coli strain involved. We note that this sentence has been deleted in the new version.
L 211-214 - ..."Although we do not have enough data, it could be hypothesised that platelets may represent an unavoidable barrier during E. coli bacteraemia and that the damage that occurs requires further re sistance to this barrier":.. It is not clear the author's hypothetical statement. Such requires a more careful analysis. Since there is no hard experimental evidence for this, such loose speculation is inadvisable
We noticed that the 5 clinical strains with pathogenicity are all platelet-resistant. In contrast, 3 of the 5 laboratory strains, which have never been implicated in pathology, have shown sensitivity. We believe that this remark deserves to be highlighted to draw reader attention for possible future studies. However, our results are preliminary. So, as requested, we have deleted this sentence.

Round 2
Reviewer 2 Report
I accept the introduced amendments / additions and their justification. I have no further serious comments, but I see a few more corrections that need to be made.
L - 19; 293, 298 - behaviour or behavior (?)
L- 35 - ..... "which showed the ability to bind directly to GPIbα" (should be not in italic)
L- 156 - Legend to Fig. 1. - Are the "Plt" columns really yellow?
L- 235, 237, 247, 248 and in a few other places - providing only the ATCC collection number without the species name of the microorganism is not correct.
L - 293-294 - ...."We hypothesised that this difference in behaviour could be related to colistin resistance. We did not confirm this hypothesis, because 2 colistin-susceptible strains 294 have the same profile as colistin-resistant strains". This interesting hypothesis, even if not fully supported, should be discussed in more detail in the context of the pathogenesis of E. coli infections and antibiotic therapy.
L- 308-310 - ..."On the other hand, our preliminary data ob-308 tained by studying the structure of the O antigen of two laboratory strains suggest that 309 modification of the O antigen would be responsible for this sensitivity to the bactericidal 310 activity of platelets." ....
What the authors think about the advisability of using isolated LPS preparations of different structures instead of bacteria in a similar research model. Could some of the hypotheses be supported experimentally then? Such considerations seem justified in view of the fact that many of the pathological effects during infection on immunologically competent cells (including platelets) depend on LPS released from the bacterial wall spontaneously and / or through the action of drugs and biocidal products of these cells.
Author Response
Pr Laurence Camoin-Jau
Corresponding author
Service d’Hématologie Biologique
Assistance Publique Hôpitaux de Marseille
Centre Hospitalier Universitaire Timone
164 Rue Saint Pierre
13005 Marseille
France
Courriel: Laurence.camoin@ap-hm.fr
Marseille, June 7th 2022
Dear Reviewer,
Please find enclosed our response.
L - 19; 293, 298 - behaviour or behavior (?)
As requested, this mistake was corrected.
L- 35 - ..... "which showed the ability to bind directly to GPIbα" (should be not in italic)
As requested, this mistake was corrected, line 25, page 3.
L- 156 - Legend to Fig. 1. - Are the "Plt" columns really yellow?
The “Plt “ column is really yellow
L- 235, 237, 247, 248 and in a few other places - providing only the ATCC collection number without the species name of the microorganism is not correct.
As requested, the species name of the microorganism was added.
L - 293-294 - ...."We hypothesised that this difference in behaviour could be related to colistin resistance. We did not confirm this hypothesis, because 2 colistin-susceptible strains 294 have the same profile as colistin-resistant strains". This interesting hypothesis, even if not fully supported, should be discussed in more detail in the context of the pathogenesis of E. coli infections and antibiotic therapy.
We have taken your comment into account and have added the following paragraph on line 296, page 16.
“Resistance to colistin can also be implied in generating a difference in responses to platelets, since LPS more precisely lipid A, represents the target of colistin, which is also the principal element that interacts with the platelet receptor TLR4. A possible modification of LPS can cause a defect in the interaction with platelets, which can lead to platelet non-activation and resistance to platelet peptides, especially cationic ones which share several characteristics with colistin, namely their polarity and their modes of action [42,43]. Moreover, the 5 colistin resistant strains could not activate the platelets and they are all resistant to the platelets which can constitute a problem during an E. coli infection which should not be neglected”.
L- 308-310 - ..."On the other hand, our preliminary data obtained by studying the structure of the O antigen of two laboratory strains suggest that modification of the O antigen would be responsible for this sensitivity to the bactericidal activity of platelets." ....
What the authors think about the advisability of using isolated LPS preparations of different structures instead of bacteria in a similar research model. Could some of the hypotheses be supported experimentally then? Such considerations seem justified in view of the fact that many of the pathological effects during infection on immunologically competent cells (including platelets) depend on LPS released from the bacterial wall spontaneously and / or through the action of drugs and biocidal products of these cells.
We share your opinion. The use of isolated LPS preparations of different structure s could indeed allow us to better understand the observed differences in behavior. This could be the subject of other work.
